# Can primary care research be conducted more efficiently using routinely reported practice-level data: a cluster randomised controlled trial conducted in England?

Peter S Blair [ID],[1] Jenny Ingram [ID],[1] Clare Clement [ID],[2] Grace Young,[2] Penny Seume,[3] Jodi Taylor,[2] Christie Cabral,[3] Patricia Jane Lucas,[4] Elizabeth Beech,[5] Jeremy Horwood,[3] Padraig Dixon [ID],[6] Martin C Gulliford [ID],[7] Nick Francis [ID],[8] Sam T Creavin,[3] Athene Lane,[2] Scott Bevan,[2] Alastair D Hay[3]

For numbered affiliations see end of article.

**Correspondence to**
Professor Peter S Blair, 1-5 Whiteladies Road, Clifton, Bristol, BS8 1NU, UK; p.s.blair@bris.ac.uk

## ABSTRACT

**Objectives** Conducting randomised controlled trials (RCTs) in primary care is challenging; recruiting patients during time-limited or remote consultations can increase selection bias and physical access to patients' notes is costly and time-consuming. We investigated barriers and facilitators to running a more efficient design.

**Design** An RCT aiming to reduce antibiotic prescribing among children presenting with acute cough and a respiratory tract infection (RTI) with a clinician-focused intervention, embedded at the practice level. By using aggregate level, routinely collected data for the coprimary outcomes, we removed the need to recruit individual participants.

**Setting** Primary care.

**Participants** Baseline data from general practitioner practices and interviews with individuals from Clinical Research Networks (CRNs) in England who helped recruit practices and Clinical Commission Groups (CCGs) who collected outcome data.

**Intervention** The intervention included: (1) explicit elicitation of parental concerns, (2) a prognostic algorithm to identify children at low risk of hospitalisation and (3) provision of a printout for carers including safety-netting advice.

**Coprimary outcomes** For 0–9 years old—(1) Dispensing data for amoxicillin and macrolide antibiotics and (2) hospital admission rate for RTI.

**Results** We recruited 294 of the intended 310 practices (95%) representing 336 496 registered 0–9 years old (5% of all 0–9 years old children). Included practices were slightly larger, had slightly lower baseline prescribing rates and were located in more deprived areas reflecting the national distribution. Engagement with CCGs and their understanding of their role in this research was variable. Engagement with CRNs and installation of the intervention was straight-forward although the impact of updates to practice IT systems and lack of familiarity required extended support in some practices. Data on the coprimary outcomes were almost 100%.

## STRENGTHS AND LIMITATIONS OF THIS STUDY

⇒ Using routinely collected data in primary care at the practice level removes the burden of individual patient recruitment and potential for selection bias.

⇒ Using the National Institute for Health Research Clinical Research Network across England and Clinical Commissioning Groups (CCGs) allowed recruitment of practices with a broader research experience.

⇒ Embedding the intervention within practice systems utilises existing patient data.

⇒ Engagement with third parties (such as the CCGs) to collect primary outcome data adds another layer of administrative burden.

⇒ Data collected at the practice level are limited (eg, absence of denominator data such as patients consulting for different conditions) so are only viable if suitable proxy markers can be found for the outcomes of interest.

**Conclusions** The infrastructure for trials at the practice level using routinely collected data for primary outcomes is viable in England and should be promoted for primary care research where appropriate.

**Trial registration number** ISRCTN11405239.

## INTRODUCTION

Conducting randomised controlled trials (RCTs) in primary care is essential for the development of a robust evidence base to improve the treatment, health and well-being of patients. However, it is a difficult environment in which to conduct research. In England, general practices are already divided into those who are willing to conduct research and those who are not which challenges the external validity of any successful trial when rolling out an intervention.[1]

Effective recruitment strategies generally require practitioner involvement[2] but this is difficult in the time allowed for consultations and can lead to the exclusion of patients for whom recruitment might be more challenging and therefore increase the risk of selection bias.[3] The time pressure can be compounded if the intervention is not fully integrated into the practice computer systems; a stand-alone tool takes time to open and close, may not draw on information already collected within the system and may be less amenable to modification on a wider scale. In primary care research, there are also difficulties in collecting patient outcomes. This is largely dependent on physical access to patient notes and is both costly and time-consuming.

Conducting trials at the practice level removes the need for clinicians to recruit individual patients and opens up the possibility of utilising routinely collected data for patient groups at each practice. Nationally collected routine data by practice and patient age are available from Clinical Commissioning Groups (CCGs)[4] and reliable data related to activity or financial transactions, such as the dispensing of medications and hospitalisation rates can, be used for primary trial outcomes. The research infrastructure in England created by the National Institute for Health Research (NIHR) could make the wider recruitment of practices viable.[5] Clinical Research Networks (CRNs) support health and care organisations to be research active[6] and can help recruit practices at a national level, including those serving diverse socioeconomic populations, improving the generalisability of findings.[7] Over 90% of the 7526 general practitioner (GP) practices in England use either the Egton Medical Information Systems (EMIS) (56%) or SystmOne (34%) and both have the facility to integrate intervention algorithms and use patient data already on the system.[8] Simpler designed studies, which place fewer demands on clinicians and practices compared with other studies, may also encourage research-naïve practices to take part in research.

Recruitment at the patient level was found to be a challenge in our feasibility study with a significant differential in the health of the patients between arms, increased consultation times due to individual recruitment and using a stand-alone intervention and costly in terms of collecting individual patient notes.[9 10] The ongoing main trial, which aims to reduce antibiotic prescribing among children presenting with an upper respiratory tract infection (RTI) and cough (The CHIldren's COugh, CHICO trial) was redesigned at the practice level.[11] We report on our experience of a simpler design; recruiting practices nationally utilising routinely collected data as the primary outcome, integrating the intervention within electronic health record systems and a light-touch approach to data collection. We also used a brief survey to find out how CRNs communicate with practices and semi-structured interviews of key individuals in CRNs and CCGs to look at the barriers and facilitators to this approach.

## METHODOLOGY

RTIs in children are common and present major resource implications for primary care.[12 13] Unnecessary use of antibiotics is associated with the development and proliferation of antimicrobial resistance. In 2016, our 5-year NIHR-funded 'TARGET' programme developed a prognostic algorithm to identify children with acute cough and RTI at very low risk of hospitalisation within 30 days and unlikely to need antibiotics.[14] The intervention included: (1) explicit elicitation of parental concerns, (2) the results of the prognostic algorithm accompanied by prescribing guidance and (3) provision of support for a no-antibiotic strategy through a printout for carers including safety netting advice.[15]

The subsequent feasibility study, recruiting at the patient level, showed prescribing reductions in both arms of the trial but also exposed the differential recruitment of significantly healthier children in the control arm. In the qualitative interviews, clinicians reported preferential recruitment of less unwell children as these were quicker to manage and therefore easier to recruit.[10]

To negate differential recruitment, and conserve resources, an 'efficient design' was proposed for the full trial. The main changes in design were: (1) Recruiting practices (with the help of CRNs and CCGs) rather than individual patients and (2) Using routinely collected data from CCGs and from national reporting systems for the primary outcome rather than directly from the practices.

1. Integrating the intervention within electronic medical records (with a triggered pop-up) rather than a stand-alone web-based tool.
2. A light-touch approach to collecting secondary data using practice champions (eliciting the help of a practice manager or someone familiar with practice systems) rather than accessing patient notes.

The CHICO RCT is an efficient, pragmatic open label, two-arm (intervention vs usual care) trial with an embedded qualitative study, with randomisation at the practice level, using routine antibiotic dispensing and hospitalisation data to assess effectiveness. The study population is children aged 0–9 years presenting with acute cough and RTI. Oral suspensions are more often given to this age group. The setting is consultations in primary care practices with prescribing clinicians in diverse regions across England. Recruitment is at the practice level, so consent is not required for individual participants. Recruitment of practices is via CCGs and by using the CRN who support patients, the public and health and care organisations across England to participate in high-quality research.

Feedback on the roles of CRNs and CCGs in recruiting practices were obtained from a short questionnaire sent to all CRNs and semistructured interviews with a convenience sample of key individuals in CRNs and CCGs. The questionnaire focused on how CRNs communicate with practices and the subsequent interviews explored these points and individual opinions of conducting efficient

design trials in primary care in greater depth. The interviews were conducted in September 2021.

Questionnaire responses were summarised in a table and pertinent comments highlighted. Interviews were audiorecorded, transcribed verbatim and analysed using a framework approach.[16] Transcripts were all read for familiarisation with the data and an initial sample of transcripts using both deductive (key issues in the interview topic guide) and inductive (derived from the data) codes. The initial codes were discussed with the wider team and a thematic framework was developed and used to code the whole of the dataset. Findings were then summarised in table format and used to inform wider understanding of the facilitators and barriers to conducting CHICO.

### Patient and public involvement

This intervention has been developed collaboratively with our parent advisory group and clinical advisory group throughout the 'TARGET' programme. Their comments and suggestions about the format of the intervention and parent/carer materials have informed the intervention and the design of the earlier feasibility study and the design of the main trial.

## RESULTS
### Recruiting practices

The sample size calculation indicated we needed 310 practices (155 intervention, 155 usual care). Between October 2018 and October 2020, we recruited 294 practices (94.8%) representing 336 496 registered patients aged 0–9 years old (5.0% of all 0–9 years old children in England).[17] Practices were recruited using all 15 CRNs in England and 47 of the 211 CCGs covering the English regions in 2019 (figure 1).[18] Table 1 gives an indication of the generalisability of the results. Over half of the practices (59%) were larger than the average list size in England, around one-third (32%) had higher prescribing rates than the national average and over a quarter of the practices (26%) were in the most deprived socioeconomic quintile reflecting the national distribution of more practices being located in urban areas.

Assuming most families live close to their GP practice, using Income Deprivation Affecting Children Index, we estimate 15% of children included in the trial lived in a deprived neighbourhood (defined as those people that are out-of-work, and those that are in work but who have low earnings). Typically, the number of registered 0–9 years old was just under 1000 per practice, staffed with a median of 6 GPs, 2 salaried nurses, 1 pharmacist and 3 locums over 1 year. The practice list size of 0–9 years old children ranged from 149 to 6969 with 64 practices (22%) having more than 1500 children registered (figure 2A). The number of amoxicillin or macrolides dispensed, over the 12-month baseline period prior to randomisation ranged from <5 to >55 per 100 patients (figure 2B) with a median of 18 items (table 1). Recruitment was planned over a 12-month period but took 24 months to complete.

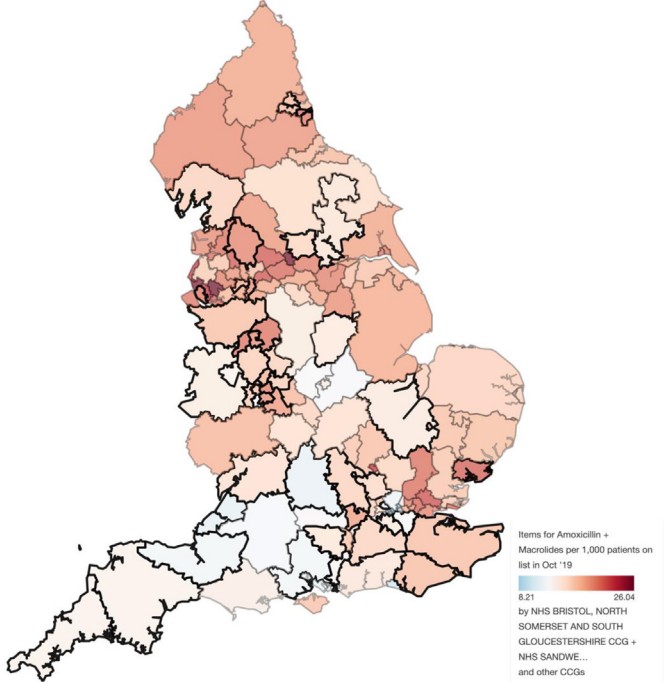

**Figure 1** The CCGs taking part in the CHICO study (bold), shaded according to the number of items of amoxicillin and macrolides, per 1000 list size at the mid point of recruitment (October 2019). Source: OpenPrescribing. net, EBM DataLab, University of Oxford, 2017. For up to date data please refer to: [https://openprescribing.net/ analyse/#org=CCG&orgIds=15E,15C,D2P2L,14Y,06H,04Y, 27D,15N,01A,05D,97R,D4U1Y,11M,D9Y0V,06N,91Q,11N, 15F,99A,14L,01K,13T,06T,05G,W2U3Z,52R,10Q,01G,05Q, 11X,M2L0M,05W,72Q,36L,05V,00P,92A,03Q&numIds= 0501013B0,5.1.5&denom=total_list_size&selectedTab=map]. CCGs, Clinical Commission Groups; CHICO, CHIldren's COugh; NHS, National Health Service.

This in part was due to first having to obtain agreement from CCGs (providing the coprimary outcomes) to participate in the study and delays due to the COVID-19 pandemic.

### Facilitators

The 15 CRNs were helpful with the recruitment of practices (table 2); presenting our study at CRN meetings and using the national CRN lead for support helped facilitate a co-ordinated national approach.

Feedback from 11 of the 15 CRNs indicated that some contact practices that opt-in to conduct research while others contact all the practices in their area; around half the practices had joined the Research Sites Initiative scheme (NIHR-funded scheme to enable research delivery) and were often the first to be contacted. Recruitment via CCGs had a wider reach of practices than via CRNs, although CCG participation in recruitment was more variable depending on capacity. Using quarterly study newsletters for practices, CRNs and CCGs with league tables monitoring levels of recruitment improved responses.

**Table 1** Practice baseline characteristics

| | n | CHICO practices (n=294) n (%) or median (IQR) |
|---|---|---|
| Total no of 0–9 year olds listed* | | 339 496 |
| Practice characteristics, using routine data sources | | |
| Above the average list size in England† | 294 | 173 (59%) |
| Median list size of 0–9 years old (IQR)* | 294 | 984 (678–1405) |
| Above the median prescribing rate in England‡ | 294 | 93 (32%) |
| Median prescribing rate (IQR)§ | 294 | 17.9 (14.3–24.6) |
| Patient characteristics, using a practice questionnaire¶ | | |
| Median # of general practitioners | 290 | 6.0 (4.0, 9.0) |
| Median # of salaried nurses | 238 | 2.0 (1.0, 4.0) |
| Median # of sessional nurses | 133 | 0.0 (0.0, 1.0) |
| Median # pharmacist independent prescribers | 190 | 1.0 (0.0, 1.0) |
| Median # of Locums in previous 12 months | 203 | 3.0 (2.0, 6.0) |
| Median distance to the nearest childrens A&E (miles) | 287 | 4.5 (2.2, 10.0) |
| IMD quintile based on practice postcode | | |
| 1 (most deprived) | | 77 (26%) |
| 2 | | 60 (20%) |
| 3 | 294 | 60 (20%) |
| 4 | | 54 (18%) |
| 5 (least deprived) | | 43 (15%) |
| Income Deprivation Affecting Children Index Score** | | |
| Median proportion (IQR) | 294 | 15% (8%, 26%) |

*Based on the mean practice list size of 0–9 years old over the 12 months prior to randomisation.
†The median practice list size of 0–9 years old in England in October 2019, the midpoint of recruitment, was 852 patients 14.
‡Based on amoxicillin and macrolide use in England CCGs (median=22) for all ages.[29]
§Based on the total number of amoxicillin/macrolide items dispensed over the 12 months prior to randomisation, divided by the practice list size.
¶As reported by the practice champion.
**The Income Deprivation Affecting Children Index measures the proportion of all children aged 0–15 living in income deprived families. It is a subset of the Income Deprivation Domain which measures the proportion of the population in an area experiencing deprivation relating to low income. The definition of low income used includes both those that are out-of-work, and those that are in work but who have low earnings (and satisfy the respective means tests).
A&E, Accident & Emergency; CCGs, Clinical Commission Groups; CHICO, CHIldren's COugh; IMD, Index of Multiple Deprivation.

## Barriers

Limiting the contact to practices that want to opt-in to research via some of the CRNs misses the opportunity of letting research-naïve practices know about light-touch efficient design studies. The level of engagement from CRNs varied slightly but much more between CCGs. Some CCGs were averse to getting involved in research or cited lack of capacity as a reason to be excluded from the research (table 2). It was difficult to know which individuals to contact and how to do so (some CCGs did not appear to be public-facing), response times were often slow, their role in research was often misunderstood, staff changes hindered communication and a number of CCGs merged during the study period. At the start of recruitment, October 2018, there were 211 CCGs across England. By the end of follow-up, September 2021, there were only 106 CCGs. While this did reduce the number

of CCG contacts required to obtain the data, it sometimes resulted in change of staff who were not familiar with the trial or its requirements. Of the 294 practices we recruited, we are aware of at least 22 practices (7%) who merged during their baseline/follow-up annual data capture. We excluded practices who anticipated a merger with another practice but had no control over this once the practice was randomised, especially in the rare instances when the merging practices were randomised to different arms of the trial. The length of time from expression of interest from practices to randomisation was longer than expected due to the delayed site agreements returned from the practices.

## Using routinely collected data for the primary outcomes

The co-primary outcomes in the trial were (1) practice dispensed prescription data for amoxicillin and macrolide

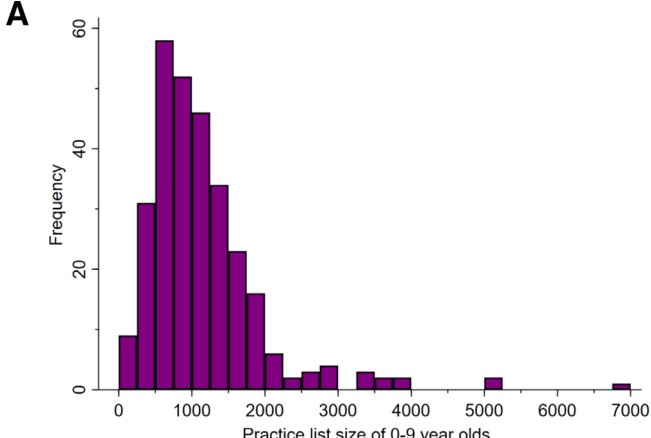

**A**

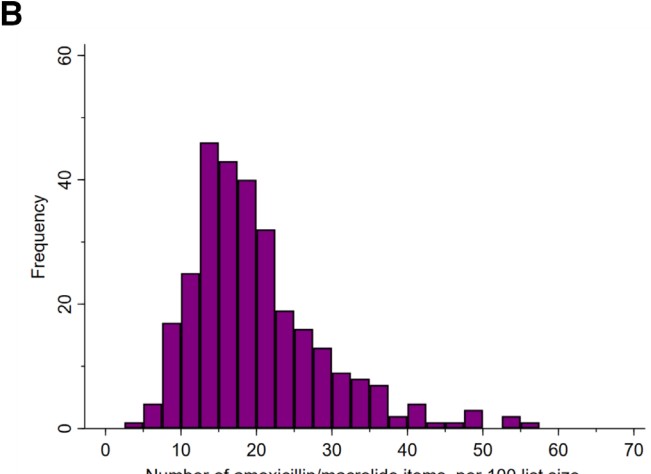

**B**

**Figure 2** (A) The distribution of practice list sizes*, of 0–9 years old, for those practices taking part in the CHICO study. *Defined for each practice as the mean list size of 0–9 years old, over the 12 months prior to randomisation (source: https://digital.nhs.uk/data-and-information/publications/statistical/patients-registered-at-a-gp-practice). (B) The distribution of practice dispensing rates* at baseline, for 0–9 years old, for those practices taking part in the CHICO study. *Defined for each practice as the number of amoxicillin or macrolides dispensed, over the 12-month baseline period prior to randomisation, divided by the practice list size. This may include multiple items per child. CHICO, CHIldren's COugh.

antibiotics for children 0–9 years and (2) hospital admission rate for RTI among 0–9 years old. The antibiotic dispensing rates were retrieved from the *NHSBSA ePACT2* reporting system for all practices in the trial and practice list size data by 5-year epoch was retrieved from National Health Service Digital and combined to create a dispensed prescription rate for amoxicillin and for macrolide antibiotics.[19] The hospital admission rates are routinely collected by all English CCGs. The ascertainment of these data was 99.7% (293/294) for antibiotic dispensing and hospitalisation rates. Data for one practice were lost due to a merger.

## Facilitators
Using aggregated data avoids the need for the consulting physician to solicit individual consent, reduces or

eliminates the risk of selection bias and removes the task of accessing individual patient notes. Routine data are often collected monthly so data completeness and quality can be monitored throughout the collection period, which was particularly important in this trial to scrutinise and report any sudden changes in hospital admission rates to the data monitoring committee. The routinely collected data can be imported to data management software, thus there is less likelihood of data entry errors and missing data are rare. The cost of research staff and time taken to collect the primary outcome data was much reduced compared with a traditional trial. In total 11/294 practices asked to be withdrawn from the study (3.7%) and 14/294 (4.8%) were lost to follow-up from the study; the main reasons cited being lack of capacity and prioritising the COVID-19 pandemic. A further advantage of collecting routine data is that withdrawal of a practice does not necessarily mean a loss of the primary outcome data for that practice (if the data are already in the public domain).

## Barriers
A potential problem with aggregate data collected from a third party is where data are suppressed, owing to a low number of events, although in this study data were collected over a 12-month period largely avoiding this problem. Some liaison was needed between the trial team and the CCGs to know exactly what data were available and in what format this could be presented. Dispensed prescription data, practice list size data and hospital admission data are reliable as they feed into financial transactions, however data reporting emergency department (ED) attendance were less so due to a limited coding set. Many ED attendances do not have a diagnosis coded, therefore, the number of attendances attributable to RTI is likely to be inaccurate.

## Integrating the intervention within the electronic medical record system
The trial only included practices using the *EMIS* system. The practice champion was given written instructions on how to instal the intervention within *EMIS*. Self-directed training materials were provided for all clinicians using the intervention. The algorithm consists of seven predictors of future hospitalisation, two of which were already in the practice systems (child age and history of asthma) and thus pulled into the algorithm automatically from patient records, and five which were entered during consultation (illness duration, raised temperature, vomiting in last 24 hours, presence of wheeze and presence of intercostal or subcostal recession).[20] Carer concerns were elicited during the consultation and formed part of a personalised leaflet generated from the system containing an easy to read 'Caring for Children with Coughs' graphic and safety-netting advice about when to seek medical care or advice.[16] Intervention use was monitored using searches on the EMIS system, run by the practice champion and shared with the research team.

**Table 2** Semistructured interviews with members of CRNs and CCG

| Question | Response |
|---|---|
| What are the facilitators to practice recruitment? | "Communicating with the practices. I think in an academic centre, if you're writing to practices, they don't really know you, do they? Whereas actually, if we put it in our local bulletins and keep encouraging practices to think about it, then it's probably a better way of increasing uptake."(P4 CCG) |
| | "There's been a nice friendship [with study team] and trying to create the right words to share with practices." [P1 CCG] |
| | "Just keep on plugging. I think quite often they're either going through a really busy time, I've contacted the wrong person or there's not enough information there for them to make a snap decision. The more information I've got and the simpler I make it, then the more likely they are to say yes or no." [P3 CRN] |
| What are the barriers to practice recruitment? | "To be honest it was a little bit more difficult to get the paperwork signed by the CCG as well within the practices, 'cause a lot of the practices we generally have a named contact, someone who already knows us and we've got relationships with…so it was a lot easier to get engagement than it was from CCGs." [P2 CRN] |
| | "I try to avoid the practice managers making the decisions if I can because they are good gatekeepers. Well our practice is very busy, that GP won't be interested when actually sometimes they quite often are." [P3 CRN] |
| Tell us about Research active and non-active (naïve) practices? | "It's usually the research active sites that get back to us but sometimes others do respond and want to take part in studies and we've got this ongoing engagement programme with all the practices in ((city)) to try and get more of them on board with research even if it's just doing simple stuff." [P5 CRN] |
| | "I would say that about a third of the practices were what I would call research naïve or inexperienced, green as in they hadn't really had a sort of established relationship with us in the past." [P2 CRN] |
| Tell us about the role of CCGs in Research? | "I mean in reality they [CCG] don't play a major part in—and never actually have a major role in research. It's not a core business of a CCG like it would be in a provider. So we don't have a research department, which is probably why the CHICO trial ended up at my door because it was about prescribing." [P4 CCG] |
| Tell us about efficient design trials? | "Yeah, go for it because once you've got the practice on board, it's almost like they don't have to do as much either, I would definitely encourage practices to take on this type of research, definitely, as long as you've got everybody on board and it all works through then yeah, there's no reason why not." [P3 CRN] |
| | "It seems quite suited to primary care because I think primary care's biggest issue is time. So, if they've got a patient in front of them, the chance of them actually getting that patient to consent is probably quite low because it's just they—that's an extra minute on a ten-minute appointment isn't it really?"(P4 CCG) |

CCG, Clinical Commission Group; CRNs, Clinical Research Networks.

## Facilitators

Installation of the intervention was relatively straight-forward for EMIS. Practice systems allow for user-friendly manipulation so interventions can be integrated and interface with system data already collected, thus negating any additional clinical workload. Screen pop-ups can also be used to notify clinicians of eligible patients to the study.

## Barriers

Familiarity with the EMIS system varied between practices thus the level of support required from the research team to help instal the intervention and download usage also varied. Provision to help instal or use third-party algorithms is not offered by system providers. EMIS upgrades to the system during the trial meant that rewriting installation instructions, resources and testing had to be carried out and fed back to the practices. For instance, the READ codes used to identify clinical terms within consultations were upgraded by EMIS in the early months of 2020 which meant the algorithm had to be amended so the intervention would function correctly. We found that our funded provision of IT support throughout the study was crucial to the smooth running of the integrated intervention. Although pop-ups can be used, some clinicians found them irritating and switched them off while other practices had so many pop-ups (especially during the COVID-19 pandemic) that the CHICO pop-up was often obscured.

### A light-touch approach to data collection

Clinicians were not required to provide any data about individual participants (apart from reporting serious adverse events (SAEs)), those in the intervention group were asked to familiarise themselves with the tool and use

it, while those in the control arm were asked to provide usual care. Data collection from the practices directly was limited to short baseline and follow-up questionnaires. Data on intervention usage were downloaded from *EMIS* and the co-primary outcome downloads from ePACT2 and the relevant CCGs.

### Facilitators

The light touch approach reduced the time needed during consultation to record information, the trawling of patient notes and provided a more objective data resource downloaded from the system rather than from individual input. Practice champions, familiar with practice systems, played an important role in obtaining the required data. Baseline questionnaires were received from 294/294 practices (100%), follow-up questionnaires were received from 265/294 practices (90%) while intervention usage was collected from 116/144 of the intervention practices (81%), indicating the data burden was not too onerous.

### Barriers

Conversely, this approach reduced the level of interpretation that can be gleaned from the data. Removing patient level recruitment results in a loss of denominator data, in our case not knowing how many individual children consulted for RTI and cough. As a proxy we used the number of children registered at each practice. Given the diversity of the practices included in the trial we are assuming that those children taking part in the trial were no different from children in the general population, but we have no way of testing this assumption. We also lost detail that may be taken from the consultation of whether antibiotics were prescribed immediately or delayed relying instead on the number of antibiotics being dispensed. The lack of contact with usual care practices (apart from reminders to provide SAE reports) runs the risk of practices wanting to withdraw from the study, especially with changes of staff. Consideration also needs to be given to the ethical implications of not seeking consent from individual patients.

### DISCUSSION

The NIHR is keen to see the design, development and delivery of more efficient, faster, innovative studies which provide robust evidence to inform clinical practice and policy.[21] The CHICO study demonstrates that an efficiently designed practice-level large trial in primary care using routinely collected data is feasible and potentially good value for money. The average cost of an HTA RCT was £1.25 million in 2019/2020,[22] whereas the cost of the CHICO RCT, which included over 300 000 children (5% of the entire national 0–9 years old population) and 4% of all GP practices in England, was below £1 million. Using routinely collected data as the primary outcome reduces problems with missing data while removing the burden of patient recruitment and focusing the clinician's time

on using the intervention reflects real life practice. These findings are pertinent to the healthcare system in England but might lend themselves to similar primary care networks in other countries.

Integrating the intervention within the practice system both exploits the data already available and adds to the patient's record avoiding duplicating of effort and saves time. Primary care practices are often very busy and the average length of face-to-face consultation in the UK is around 10 min; less than half the time given to patients in, for example, Sweden and the USA.[23] Growing demands on primary care services have also led to policy-makers promoting telephone and video consultations, even before the COVID-19 pandemic, and these sometimes do not lend themselves to enlisting patients in research.[24] Reducing the research burden for participants and clinicians is always desirable, but particularly so given the increasing time constraints in primary care. This light-touch approach may also be more appealing for practices who maybe research naïve; investing in different recruitment strategies[25] using existing networks could potentially yield a more representative sample than previous trials in primary care from which to generalise any findings. The design of the CHICO trial retrospectively scored highly in each domain of the (PRagmatic Explanatory Continuum Indicator Summary) tool[26] suggesting it is a pragmatic randomised trial focusing on delivery in the 'real world' rather than providing the best chance to demonstrate a beneficial effect in an idealised setting.

Efficient design studies which use routine data and recruit at practice level face their own challenges. First, this approach would not be suitable if individual patient consent is required or for trials using a patient reported outcome as the primary outcome, as these are not widely collected in routine care. Furthermore, the number of CCGs in England almost halved during the study period due to organisational restructuring which made it difficult to administer the trial. From March 2022 Integrated Care Systems will replace CCGs and this will provide new bureaucracy for researchers to develop relationships with. There is a lack of uniformity when approaching these commissioning groups and their role in research needs clarifying.[27] Using a convenience sample the qualitative interviews have limited insight but suggest research needs to be higher on the agenda. CRNs are more research-focused and can help with recruitment although adoption of the study needs to be made more explicit and a consistent national approach to include research naïve practices needs to be adopted within this network.

If practices are being used as the unit of analysis, then the commitment to research they have signed up for needs to be strengthened. Around 2.5% practices close or merge each year in England while some new practices open and recognition of the current research portfolio needs to be part of this process.[28] We were also surprised by the wide variability in practice list size, 8% of practices in the study having three sites or more. This has implications for future trials in terms of factoring in variable list sizes

for sample size calculations and checking that multiple sites use the same electronic record systems. A light touch approach is only viable if suitable primary outcomes can be identified but makes fidelity more difficult to measure. Losing denominator data such as patients consulting for different conditions will depend on the hypothesis being tested and needs to strike a balance between accuracy of what you are trying to measure and whether a proxy marker will deliver the population impact of the intervention. Using the patient list size of 0–9 years old as a proxy denominator instead of those consulting for RTI accounts for the variability between the size of different practices but not necessarily the disease burden.

If one of the main intentions of primary care research is to provide the clinician with better tools, then we need to work more closely with those who supply the toolbox (IT system providers) and the third parties who provide the data within which practices work. The infrastructure to conduct efficiently designed trials that do not require patient-reported outcomes in England is potentially viable but does require more investment in time and effort to make recruitment of practices and data collection more accessible to researchers.

**Author affiliations**
[1]Centre for Academic Child Health, University of Bristol, Bristol, UK
[2]Bristol Trials Centre, University of Bristol, Bristol, UK
[3]Centre for Academic Primary Care, University of Bristol, Bristol, UK
[4]School for Policy Studies, University of Bristol, Bristol, UK
[5]NHS Improvement South West, NHS England, London, UK
[6]Oxford University, Oxford, UK
[7]King's College London, London, UK
[8]School of Primary Care Population Sciences and Medical Education, University of Southampton, Southampton, UK

**Acknowledgements** This study was designed and delivered in collaboration with the Bristol Randomised Trials Collaboration (BRTC), part of the Bristol Trials Centre, is in receipt of National Institute for Health Research CTU support funding. The University of Bristol is acting as the sponsor for this trial and the trial is hosted by the NHS Bristol, North Somerset and South Gloucestershire Clinical Commissioning Group (CCG). The authors would like to thank all General practices, CCGs and CRNs for their involvement in CHICO. The authors would also like to thank the members of the TSC and DMC.

**Contributors** ADH, PSB, PJL, NF and JI were responsible for developing the research questions. PSB, ADH, JI, PJL, CCa, CCl, EB, MCG, JH, STC, AL and NF and are responsible for the study design and collection of data. PS and SB are responsible for study management and coordination. GY, PD and CCl are responsible for the analysis of the data. PSB drafted the paper and is guarantor of the data. All authors read, commented on and approved the final manuscript.

**Funding** This research is funded by the National Institute for Health Research (NIHR) Health Technology Assessment (HTA) programme (funder ref: 16/31/98).

**Disclaimer** The views expressed are those of the authors and not necessarily those of the NIHR or the Department of Health and Social Care.

**Map disclaimer** The inclusion of any map (including the depiction of any boundaries therein), or of any geographic or locational reference, does not imply the expression of any opinion whatsoever on the part of BMJ concerning the legal status of any country, territory, jurisdiction or area or of its authorities. Any such expression remains solely that of the relevant source and is not endorsed by BMJ. Maps are provided without any warranty of any kind, either express or implied.

**Competing interests** None declared.

**Patient and public involvement** Patients and/or the public were involved in the design, or conduct, or reporting, or dissemination plans of this research. Refer to the Methods section for further details.

**Patient consent for publication** Not applicable.

**Ethics approval** This study involves human participants and was approved by London-Camden and Kings Cross Research Ethics Committee (ref:18/LO/0345). Participants gave informed consent to participate in the study before taking part.

**Provenance and peer review** Not commissioned; externally peer reviewed.

**Data availability statement** Data may be obtained from a third party and are not publicly available. The data are obtained from a third party (Clinical Commissioning Groups and Public Health England) and are not publicly available.

**ORCID iDs**
Peter S Blair http://orcid.org/0000-0002-7832-8087
Jenny Ingram http://orcid.org/0000-0003-2366-008X
Clare Clement http://orcid.org/0000-0002-5555-433X
Padraig Dixon http://orcid.org/0000-0001-5285-409X
Martin C Gulliford http://orcid.org/0000-0003-1898-9075
Nick Francis http://orcid.org/0000-0001-8939-7312

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
