## [Reviewer comments · BMJ Open]

ARTICLE DETAILS

TITLE (PROVISIONAL)	Can primary care research be conducted more efficiently using routinely reported practice-level data: a cluster randomised controlled trial conducted in England?
AUTHORS	Blair, Peter; Ingram, Jenny; Clement, Clare; Young, Grace; Seume, Penny; Taylor, Jodi; Cabral, Christie; Lucas, Patricia; Beech, Elizabeth; Horwood, Jeremy; Dixon, Pdraig; Gulliford, Martin; Francis, Nick; Creavin, Sam T; Lane, Athene; Bevan, Scott; Hay, Alastair

VERSION 1 – REVIEW

REVIEWER	Hafiz, Nashid The University of Sydney, Faculty of Medicine and Health
REVIEW RETURNED	01-Mar-2022

GENERAL COMMENTS	No further comments
---------------------

REVIEWER	Cole, Allison University of Washington, Family Medicine
REVIEW RETURNED	02-Mar-2022

GENERAL COMMENTS	This is a well written manuscript that addresses an important topic - feasible, practical research in real-world primary care settings. The authors miss an opportunity to ground their findings in the deep literature of pragmatic clinical research and discuss the context of practice-based research networks internationally. With the addition of this context, the manuscript would make a valuable contribution to our understanding of this topic. Discussion of the PRECIS model is missing and should be incorporated.
--

REVIEWER	Sidani, S Ryerson University
REVIEW RETURNED	28-Mar-2022

GENERAL COMMENTS	General comments: The general topic is of relevance. However, it is a bit challenging to get a good sense of what the authors want to relay. The organization of the content is not easy to follow. The consistency across sections is not always maintained. The authors are encouraged to clarify their ideas, to review the literature on cluster trials, and to determine if they want to frame this paper as a report of their experience in conducting the efficient trial or the results of the interviews that aimed to identify barriers and facilitators, and reorganize the paper accordingly. Specific comments (by section):
---

	Introduction: The authors introduce 'trials at the practice level' as potentially efficient designs. These appear comparable to cluster trials. It would be important to clarify if the interest is in cluster trials, which are recommended for implementation research. Different types of cluster trials are available, and their strengths and limitations have been discussed in the literature. The authors can review pertinent literature to strengthen the argument and/or to distinguish their design. The purpose of this paper is not explicitly stated: Is the goal to search for a design, or to report on the authors' experience with efficient designs, which seems to be the case. Methods: The first two paragraphs report about the authors' previous experience with the original RCT, which should be described in more detail in the introductory section to justify the search for an efficient design. It is essential to explain who 'proposed the efficient design' and on what basis. Was relevant methodological literature consulted to inform the choice of design and methods? What was the aim of the interviews? When were they held? How adequate is the sample size in light of recommendations to have at least 12 participants to reach information saturation in qualitative studies? It seems that the aim of the interview was to understand facilitators and barriers to conducting the efficient trial, as mentioned in the methods section. However, this aim was not made explicit in the introductory section. Results: The first sub-section describes the characteristics of the recruited practice – it is unclear how these results align with the aim of the interviews. How was 'feedback' from CRNs obtained and analyzed? Overall, the results as presented, reflect the authors' account of their experience supported by selected data collected during the trial and through the interviews. Many points are shared but not clearly explained for researchers, outside the trial, can understand and appreciate.
--	--

VERSION 1 – AUTHOR RESPONSE

Reviewer: 1

Ms. Nashid Hafiz, The University of Sydney

Comments to the Author:

No further comments

Response: Thanks for you time in reading our manuscript

Reviewer: 2

Dr. Allison Cole, University of Washington

Comments to the Author:

This is a well written manuscript that addresses an important topic - feasible, practical research in real-world primary care settings. The authors miss an opportunity to ground their findings in the deep

literature of pragmatic clinical research and discuss the context of practice-based research networks internationally. With the addition of this context, the manuscript would make a valuable contribution to our understanding of this topic. Discussion of the PRECIS model is missing and should be incorporated.

Response: Thanks for your supportive comments and steer towards the PRECIS tool. We agree that this article focuses on research and primary care networks in England; what we gain in the detail of this for future researchers in the UK is not as applicable internationally and would need a much larger manuscript to do that justice. We have added this limitation in the discussion

These findings are pertinent to the health care system in England but might lend themselves to similar primary care networks in other countries.

The study design scores high on all 9 domains of the PRECIS-2 tool. Again the word limitations restrict us from going into too much detail although we have added in the discussion the sentence:

The design of the CHICO trial retrospectively scored highly in each domain of the PRECIS-

2 ((PRagmatic Explanatory Continuum Indicator Summary) tool²⁶ suggesting it

is a pragmatic randomised trial focusing on delivery in the “real world” rather than providing the best chance to demonstrate a beneficial effect in an idealised setting.

26 Loudon K, Treweek S, Sullivan F, Donnan P, Thorpe KE, Zwarenstein M. The PRECIS-2 tool: designing trials that are fit for purpose. *BMJ*. 2015 May 8;350:h2147

Reviewer: 3

Dr. S Sidani, Ryerson University

Comments to the Author:

General comments:

The general topic is of relevance. However, it is a bit challenging to get a good sense of what the authors want to relay. The organization of the content is not easy to follow. The consistency across sections is not always maintained.

The authors are encouraged to clarify their ideas, to review the literature on cluster trials, and to determine if they want to frame this paper as a report of their experience in conducting the efficient trial or the results of the interviews that aimed to identify barriers and facilitators, and reorganize the paper accordingly.

Response: We take on board this reviewer’s comments and have added more clarity of what we want to relay. We are not trying to review the literature on cluster trials but recount our experiences of the barriers and facilitators of conducting an efficient trial in a primary care setting with a cluster randomised design. The efficiency gained is not so much basing the design at the practice level but rather utilising routine data at the practice level collected by third parties (CCGs) and our experience of this. The qualitative interviews with those based at these organisation are used to supplement and deepen explanation of our experiences. Below we set out the changes we have made for each specific comment.

Specific comments (by section):

Introduction:

The authors introduce ‘trials at the practice level’ as potentially efficient designs. These appear comparable to cluster trials. It would be important to clarify if the interest is in cluster trials, which are recommended for implementation research. Different types of cluster trials are available, and their strengths and limitations have been discussed in the literature. The authors can review pertinent literature to strengthen the argument and/or to distinguish their design.

The purpose of this paper is not explicitly stated: Is the goal to search for a design, or to report on the authors’ experience with efficient designs, which seems to be the case.

Response: We have reorganised both the abstract and the introduction to be a little more explicit that our aim is to report on the experience of this study design in a primary care setting. In the abstract we state:

“Design: An RCT aiming to reduce antibiotic prescribing among children presenting with acute cough and a respiratory tract infection with a clinician-focused intervention, embedded at the practice-level. By using aggregate-level, routinely collected data for the co-primary outcomes, we removed the need to recruit individual participants.”

And in the introduction we state:

‘Conducting trials at the practice level removes the need for clinicians to recruit individual patients and opens up the possibility of utilising routinely collected data for patient groups at each practice. Nationally collected routine data by practice and patient age are available from Clinical Commissioning Groups (CCGs)⁴ and reliable data related to activity or financial transactions, such as the dispensing of medications and hospitalisation rates, can be used for primary trial outcomes.’

And

‘Recruitment at the patient-level was found to be a challenge in our feasibility study with a significant differential in the health of the patients between arms, increased consultation times due to individual recruitment and using a stand-alone intervention and costly in terms of collecting individual patient notes.^{9,10} The on-going main trial, which aims to reduce antibiotic prescribing among children presenting with an upper respiratory tract infection and cough (The CHildren’s COugh or CHICO trial) was redesigned at the practice level.¹¹ We report on our experience of a simpler design; recruiting practices nationally, utilising routinely collected data as the primary outcome, integrating the intervention within electronic health record systems and a light-touch approach to data collection.’

Methods:

The first two paragraphs report about the authors’ previous experience with the original RCT, which should be described in more detail in the introductory section to justify the search for an efficient design.

It is essential to explain who ‘proposed the efficient design’ and on what basis. Was relevant methodological literature consulted to inform the choice of design and methods?

What was the aim of the interviews? When were they held? How adequate is the sample size in light of recommendations to have at least 12 participants to reach information saturation in qualitative studies?

It seems that the aim of the interview was to understand facilitators and barriers to conducting the efficient trial, as mentioned in the methods section. However, this aim was not made explicit in the introductory section.

Response: Our response above gives more detail in the introduction to justify the re-design in terms of why we chose it (differential recruitment) and the advantages of a more efficient design (avoiding individual recruitment and the availability of routine data collected at the practice level). We have expanded the methods section to include:

‘Feedback on the roles of CRNs and CCGs in recruiting practices were obtained from a short questionnaire sent to all CRNs and semi-structured interviews with a convenience sample of key individuals in CRNs and CCGs. The questionnaire focussed on how CRNs communicate with practices and the subsequent interviews explored these points and individual opinions of conducting efficient design trials in primary care in greater depth. The interviews were conducted in September 2021. Questionnaire responses were summarized in a table and pertinent comments highlighted.’ Saturation of topics discussed in the interviews was not reached as this was a convenience sample to supplement the questionnaires. We have added an aim of the questionnaires and interviews to the introduction.

'We also used a brief survey to find out how CRNs communicate with practices and semi-structured interviews of key individuals in CRNs and CCGs to look at the barriers and facilitators to this approach.'

And raised the limited insight of a convenience sample in the discussion

Using a convenience sample the qualitative interviews have limited insight but suggest research needs to be higher on the agenda.

Results:

The first sub-section describes the characteristics of the recruited practice – it is unclear how these results align with the aim of the interviews.

How was 'feedback' from CRNs obtained and analyzed?

Overall, the results as presented, reflect the authors' account of their experience supported by selected data collected during the trial and through the interviews. Many points are shared but not clearly explained for researchers, outside the trial, can understand and appreciate.

Response: The first subsection gives the reader an idea of the generalisability of the study (eg reflecting the tendency for proportionally more practices to be located in the most deprived socio-economic quintile) and future researchers some idea of what to expect in terms of the characteristics of the practices under study. We have now stated:

Table 1 gives an indication of the generalisability of the results.

The 'feedback' is now described in the introduction and methods and reported more clearly in the results section.

Feedback from 11 of the 15 CRNs indicated that some contact practices that opt-in to conduct research whilst others contact all the practices in their area; around half the practices had joined the Research Sites Initiative scheme (NIHR funded scheme to enable research delivery) and were often the first to be contacted.

VERSION 2 – REVIEW

REVIEWER	Cole, Allison University of Washington, Family Medicine
REVIEW RETURNED	06-May-2022
GENERAL COMMENTS	Well written and revisions have addressed concerns of reviewers.